# LOCAL REWEIGHTING FOR ADVERSARIAL TRAINING

## ABSTRACT

*Instances-reweighted adversarial training* (IRAT) can significantly boost the robustness of trained models, where data being less/more vulnerable to the *given* attack are assigned smaller/larger weights during training. However, when tested on attacks *different from* the given attack simulated in training, the robustness may drop significantly (e.g., even worse than no reweighting). In this paper, we study this problem and propose our solution—*locally reweighted adversarial training* (LRAT). The rationale behind IRAT is that we do not need to pay much attention to an instance that is already safe under the attack. We argue that the safeness should be *attack-dependent*, so that for the same instance, its weight can change given different attacks based on the same model. Thus, if the attack simulated in training is *mis-specified*, the weights of IRAT are misleading. To this end, LRAT *pairs* each instance with its adversarial variants and performs *local reweighting inside each pair*, while performing *no global reweighting*—the rationale is to fit the instance itself if it is immune to the attack, but not to skip the pair, in order to *passively* defend different attacks in future. Experiments show that LRAT works better than both IRAT (i.e., global reweighting) and the standard AT (i.e., no reweighting) when trained with an attack and tested on different attacks.

## 1 INTRODUCTION

A growing body of research shows that neural networks are vulnerable to adversarial examples, i.e., test inputs that are modified slightly yet strategically to cause misclassification (Carlini & Wagner, 2017a; Finlayson et al., 2019; Kurakin et al., 2017; Nguyen et al., 2015; Szegedy et al., 2014; Wang et al., 2019; Zhang et al., 2020b). It is crucial to train a robust neural network to defend against such examples for security-critical computer vision systems, such as autonomous driving and medical diagnostics (Chen et al., 2015; Ma et al., 2021; Miyato et al., 2017; Nguyen et al., 2015; Szegedy et al., 2014). To mitigate this issue, adversarial training methods have been proposed in recent years (Balunovic & Vechev, 2019; Goodfellow et al., 2015; Madry et al., 2018; Shafahi et al., 2020; Wang et al., 2020a; Wu et al., 2020). By injecting adversarial examples into training data, adversarial training methods seek to train an adversarial-robust deep network whose predictions are locally invariant in a small neighborhood of its inputs (Bai et al., 2019; He et al., 2018; Papernot et al., 2016; Raghunathan et al., 2020; Tsipras et al., 2019; Yang et al., 2020).

Due to the diversity and complexity of adversarial data, over-parameterized deep networks have insufficient model capacity in *adversarial training* (AT) (Zhang et al., 2021). To obtain a robust model given fixed model capacity, Zhang et al. (2021) suggest that we do not need to pay much attention to an instance that is already safe under the attack, and propose *instance-reweighted adversarial training* (IRAT), which performs *global reweighting* with a *given* attack. To identify safe/non-robustness instances, they propose a *geometric distance* (GD) between natural data points and current class boundary. Instances being closer to/farther from the class boundary is more/less vulnerable to the given attack, and should be assigned larger/smaller weights during AT. This *geometry-aware IRAT* (GAIRAT) significantly boosts the robustness of the trained models when facing the given attack.

However, when tested on attacks that are *different from* the given attack simulated in IRAT, the robustness of IRAT drops significantly (e.g., even worse than no reweighting). First, we find that a large number of instances are actually *overlooked* during IRAT. Figure 1(a) shows that, for approximately four-fifths of the instances, their corresponding adversarial variants are assigned very low weights (less than 0.2). Second, we find that the robustness of the IRAT-trained classifier *drops significantly* when facing an *unseen* attack. Figures 1(b) and 1(c) show that the classifier trained by

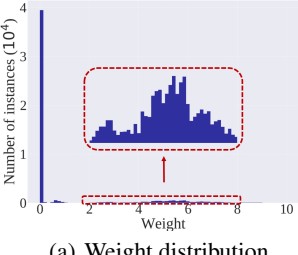
(a) Weight distribution

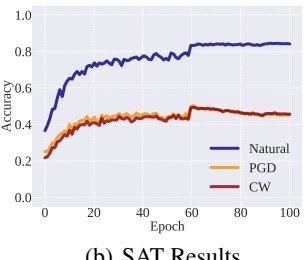
(b) SAT Results

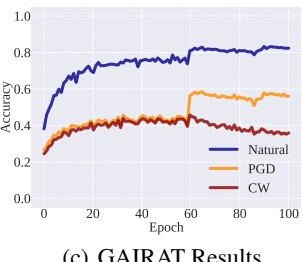
(c) GAIRAT Results

Figure 1: The extreme reweighting in GAIRAT results in a significant decrease when facing an unseen attack. The subfigure (a) illustrates the frequency distribution of weights in the GAIRAT model on the CIFAR-10 training set, where approximately four-fifths of the instances are assigned very low weights (less than 0.2). The subfigure (b) and (c) illustrate the performance on the GAIRAT trained model and the AT trained model under different attacks, which show that GAIRAT does improve the robustness when attacked by PGD (used during training), but reduces the robustness when attacked by CW (unseen during training).

GAIRAT (with *projected gradient descent* (PGD) attack (Madry et al., 2018)) has lower robustness when attacked by the unseen Carlini and Wagner attack (CW) (Carlini & Wagner, 2017b) compared with the robustness of *standard adversarial training* (SAT) (Madry et al., 2018).

Since the robustness of a model is not determined by any single attack, the improved performance for a kind of given attack is not sufficient to justify robustness. As shown in Figure 2, the ranking of attack intensity varies with different datasets, adversarial training methods and so on, which means there is not the strongest attack in all situations. For another, previous studies of the adversarial detection (Lee et al., 2018; Ma et al., 2018; Gao et al., 2021) show that the stronger adversarial examples for robust classifiers to predict, the easier for the adversarial detectors to detect. This means improving performance not only on stronger attacks, but also on weaker ones, as some of attackers may choose to use weaker attacks to escape from the adversarial detection. Hence, robustness needs the network to consider a variety of potential attacks.

In this paper, we investigate the reasons behind the failure of IRAT on unseen attacks. Unlike the common scenario of the classification problem where the training and testing data are fixed, there are different adversarial variants for the same instance in AT, e.g., PGD-based or CW-based adversarial variants. A natural question comes with this—whether there are inconsistent vulnerabilities in the view of different attacks? The answer is *affirmative*. Figure 3 visualizes this phenomenon using t-SNE (Van der Maaten & Hinton, 2008). The four subfigures in Figure 3 visualize inconsistent vulnerable instances in different views using t-SNE. The red dots in all subfigures represent consistently vulnerable instances between different views, while the blue dots represent the inconsistent vulnerable instances. From the SAT-trained classifier (Figure 3(a)-3(d)), we can clearly see that a large number of vulnerable instances are inconsistent (the blue dots dominate in all the subfigures).

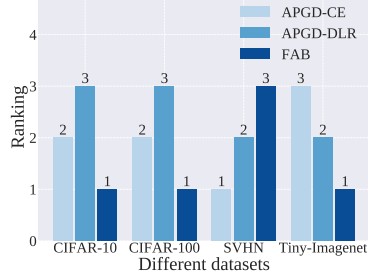

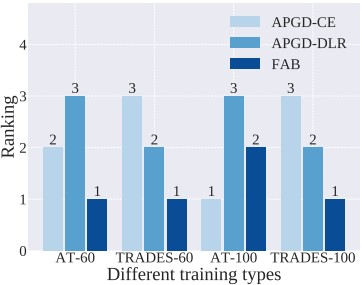

Figure 2: The ranking of attack intensity on different datasets. AT (Madry et al., 2018) and TRADES (Zhang et al., 2019) are different adversarial trianing methods. AT-60 and TRADES-60 denote the early stop models at the epoch 60.

Given the above investigation, we argue that the safeness of instances is *attack-dependent*, that is, for the same instance, its

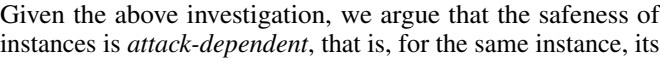

weight can change given different attacks based on the same model. Thus, if the attack simulated in training is *mis-specified*, the weights of IRAT are misleading. In order to ameliorate this pessimism of IRAT, we propose our solution—*locally reweighted adversarial training* (LRAT). As shown in Figure 4, LRAT *pairs* each instance with its adversarial variants and performs *local reweighting*

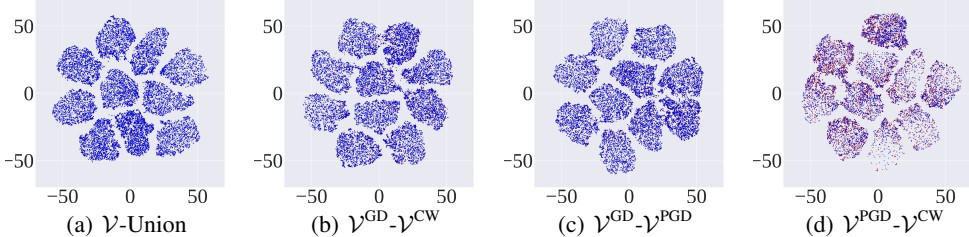

(a) $\mathcal{V}$-Union     (b) $\mathcal{V}^{\mathrm{GD}}$-$\mathcal{V}^{\mathrm{CW}}$     (c) $\mathcal{V}^{\mathrm{GD}}$-$\mathcal{V}^{\mathrm{PGD}}$     (d) $\mathcal{V}^{\mathrm{PGD}}$-$\mathcal{V}^{\mathrm{CW}}$

Figure 3: Visualization of inconsistent vulnerable instances in different views using t-SNE. The dots in subfigures are outputs of the last layers in SAT-trained classifier (subfigures (a)-(d)). Classifier inputs are natural data points in the CIFAR-10 training set. The dots represent the vulnerable instances (top-20%). The red dots represent consistently vulnerable instances, while the blue dots represent the inconsistent vulnerable instances in the view of $\mathcal{V}^{\mathrm{GD}}$, $\mathcal{V}^{\mathrm{PGD}}$ and $\mathcal{V}^{\mathrm{CW}}$. $\mathcal{V}^{\mathrm{GD}}$ is the vulnerability w.r.t. *geometry distance* (GD) defined by Zhang et al. (2021). $\mathcal{V}^{\mathrm{PGD}}$ and $\mathcal{V}^{\mathrm{CW}}$ are the vulnerability measurement function w.r.t. PGD and CW, respectively, which is defined in Eq. (5) and Eq. (6). We can clearly see that there exist abundant blue dots but only a few red dots, which means a large number of inconsistent vulnerable instances in different views.

*inside each pair*, while performing *no global reweighting*. The rationale of LRAT is to fit the instance itself if it is immune to the attack, and in order to *passively* defend different attacks in future, LRAT does not skip the pair.

In terms of the common attribute of adversarial attacks that misleading the classifier with a low probability of predicting the correct label and a high probability of predicting the false label (Madry et al., 2018; Carlini & Wagner, 2017b), we propose a general reweighting strategy that is applicable to various representatively attacks. Our experimental results show that compared with AT, LRAT can achieve better results on various simulated attacks. Other than previous global reweighting training that GAIRAT, LRAT can boost the performance of AT on PGD with no decline on other attacks.

## 2 ADVERSARIAL TRAINING

In this section, we briefly review existing adversarial training methods (Madry et al., 2018; Zhang et al., 2021). Let $(\mathcal{X}, d_\infty)$ be the input feature space $\mathcal{X}$ with a metric $d_\infty(\boldsymbol{x}, \boldsymbol{x}') = \|\boldsymbol{x} - \boldsymbol{x}'\|_\infty$, and $\mathcal{B}_\epsilon[\boldsymbol{x}] = \{\boldsymbol{x}' \mid d_\infty(\boldsymbol{x}, \boldsymbol{x}') \leq \epsilon\}$ be the closed ball of radius $\epsilon > 0$ centered at $\boldsymbol{x}$ in $\mathcal{X}$. The dataset $S = \{\boldsymbol{x}_i, y_i\}_{i=1}^n$, where $\boldsymbol{x}_i \in \mathcal{X}, y_i \in \mathcal{Y} = \{0, 1, \ldots, K-1\}$. We use $f_\theta(\boldsymbol{x})$ to denote a deep neural network parameterized by $\theta$. Specifically, $f_\theta(\boldsymbol{x})$ predicts the label of an input instance $\boldsymbol{x}$ via:

$$f_\theta(\boldsymbol{x}) = \arg\max_{k \in K} \mathrm{p}_k(\boldsymbol{x}; \theta), \qquad (1)$$

where $\mathrm{p}_k(\boldsymbol{x}; \theta)$ denotes the predicted probability (softmax on logits) of $x$ belonging to class $k$.

### 2.1 STANDARD ADVERSARIAL TRAINING

The objective function of SAT (Madry et al., 2018) is

$$\min_{f_\theta \in \mathcal{F}} \frac{1}{n} \sum_{i=1}^n \ell(f_\theta(\tilde{\boldsymbol{x}}_i), y_i), \qquad (2)$$

where $\tilde{\boldsymbol{x}}_i = \arg\max_{\tilde{\boldsymbol{x}}_i \in \mathcal{B}_\epsilon[\boldsymbol{x}]} \ell(f_\theta(\tilde{\boldsymbol{x}}_i), y_i)$ and $\mathcal{F} = \{f : \mathcal{X} \to \mathcal{Y}\}$. The selected $\tilde{\boldsymbol{x}}$ is the most adversarial variant within the $\epsilon$-ball center at $\boldsymbol{x}$. The loss function $\ell : \mathbb{R}^K \times \mathcal{Y} \to \mathbb{R}$ is a composition of a base loss $\ell_B : \triangle^{K-1} \times \mathcal{Y} \to \mathbb{R}$ (e.g., the cross-entropy loss) and an inverse link function $\ell_L : \mathbb{R}^K \to \triangle^{K-1}$ (e.g., the soft-max activation), where $\triangle^{K-1}$ is the corresponding probability simplex. Namely, $\ell(f_\theta(\cdot), y) = \ell_B(\ell_L(f_\theta(\cdot)), y)$. PGD (Madry et al., 2018) is the most common approximation method for searching the most adversarial variant. Starting from $\boldsymbol{x}^{(0)} \in \mathcal{X}$, PGD (with step size $\alpha > 0$) works as follows:

$$\boldsymbol{x}^{(t+1)} = \Pi_{\mathcal{B}_\epsilon[\boldsymbol{x}^{(0)}]}(\boldsymbol{x}^{(t)} + \alpha\,\mathrm{sign}(\nabla_{\boldsymbol{x}^t}\ell(f_\theta(\boldsymbol{x}^{(t)}, y)))), t \in \mathbb{N}, \qquad (3)$$

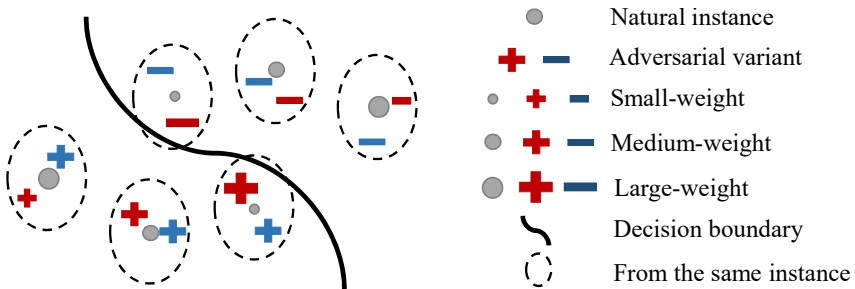

Figure 4: The illustration of LRAT. LRAT *pairs* each instance with its adversarial variants, and performs *local reweighting inside each pair* instead of global reweighting. For the same instance, there is inconsistent vulnerability in different views (the difference between red and blue). Thus, LRAT gives larger/smaller weights on the losses of adversarial variants, respectively.

where $\mathbb{N}$ is the number of iterations; $\boldsymbol{x}^{(0)}$ refers to the starting point that natural instance (or a natural instance perturbed by a small Gaussian or uniformly random noise); $y$ is the corresponding label for $\boldsymbol{x}^{(0)}$; $\boldsymbol{x}^{(t)}$ is the adversarial variant at step $t$; $\Pi_{\mathcal{B}_\epsilon[\boldsymbol{x}^{(0)}]}(\cdot)$ is the projection function that projects the adversarial variant back into the $\epsilon$-ball centered at $\boldsymbol{x}^{(0)}$ if necessary.

## 2.2 Geometry-Aware Instance-Reweighted Adversarial Training

GAIRAT is a typical IRAT proposed by Zhang et al. (2021). GAIRAT argues that natural training data farther from/close to the decision boundary are safe/non-robustness, and should be assigned with smaller/larger weights. Let $\omega(\boldsymbol{x}, y)$ be the geometry-aware weight assignment function on the loss of the adversarial variant $\tilde{\boldsymbol{x}}$, where the generation of $\tilde{\boldsymbol{x}}$ follows SAT. GAIRAT aims to

$$\min_{f_\theta \in \mathcal{F}} \frac{1}{n} \sum_{i=1}^{n} \omega(\boldsymbol{x}_i, y_i)\ell(f_\theta(\tilde{\boldsymbol{x}}_i), y_i). \tag{4}$$

Eq. (4) rescales the loss using a function $\omega(\boldsymbol{x}, y)$. This function is non-increasing w.r.t. GD, which is defined as the least steps that the PGD method needs to successfully attack the natural instances. The method then normalizes $\omega$ to ensure that $\omega(\boldsymbol{x}, y) \leq 0$ and $\frac{1}{n}\sum_{i=1}^{n}\omega(\boldsymbol{x}_i, y_i) = 1$. Finally GAIRAT employs a bootstrap period in the initial part of the training by setting $\omega(\boldsymbol{x}_i, y_i) = 1$, thereby performing regular training and ignoring the geometric-distance of input $(\boldsymbol{x}_i, y_i)$.

## 3 The Limitation of IRAT: Inconsistent Vulnerability in Different Views

The difference between Eq. (2) and Eq. (4) is the addition of the geometry-aware weight $\omega(\boldsymbol{x}, y)$. According to GAIRAT (Zhang et al., 2021), more vulnerable instances should be assigned larger weights. However, the relative vulnerability between instances may vary in different situations, such as for different adversarial variants. $\mathcal{V}$ represents the selected variable to measure the vulnerability between the classifier and adversarial variants, and the smaller $\mathcal{V}$, the more vulnerable is the instance.

For the PGD-based adversarial variant, the lower predicted probability it has on true class, the smaller $\mathcal{V}$. Here, we give the definition of $\mathcal{V}^{\mathrm{PGD}}$.

**Definition 1** (Vulnerability in the view of PGD). *In the view of PGD, the vulnerability $\mathcal{V}^{\mathrm{PGD}}$ regarding $\tilde{\boldsymbol{x}}$ (generated by PGD) is defined as*

$$\mathcal{V}^{\mathrm{PGD}}_{(\tilde{\boldsymbol{x}}, y)} := \mathrm{p}_y(\tilde{\boldsymbol{x}}), \tag{5}$$

*where $\mathrm{p}_y$ denotes the predicted probability (softmax on logits) of $\tilde{\boldsymbol{x}}$ belonging to the true class $y$.*

For the CW-based adversarial variant, the relatively higher predicted probability it has on the false class, the larger $\mathcal{V}^{\mathrm{CW}}$. Here, we give the definition of $\mathcal{V}^{\mathrm{CW}}$.

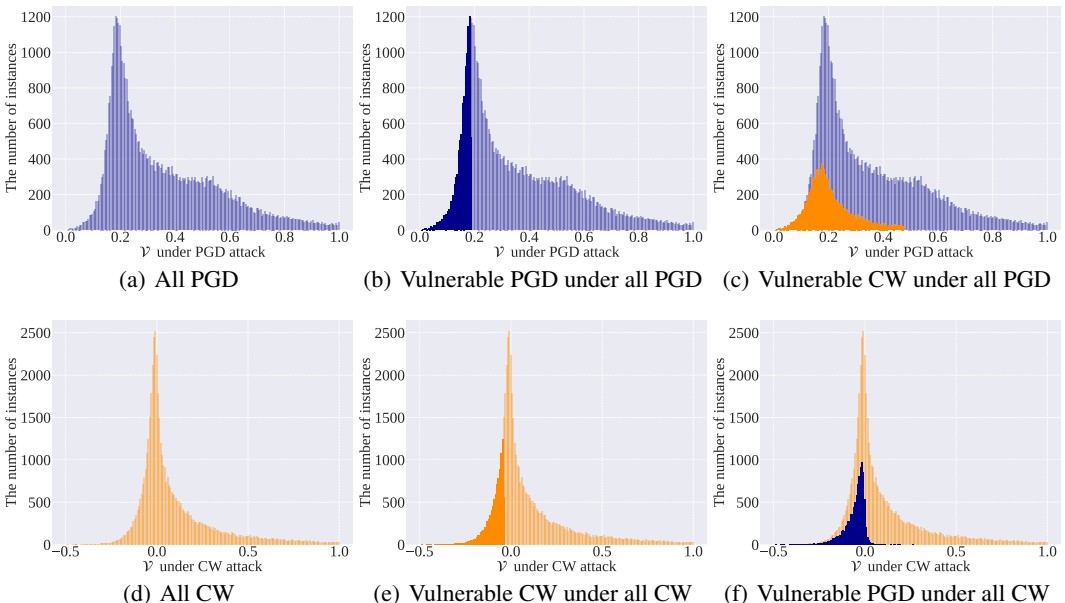

(a) All PGD    (b) Vulnerable PGD under all PGD    (c) Vulnerable CW under all PGD

(d) All CW    (e) Vulnerable CW under all CW    (f) Vulnerable PGD under all CW

Figure 5: The inconsistent vulnerability between instances about different variants. Six subfigures illustrate the frequency distribution of adversarial variants in the GAIRAT trained model on the CIFAR-10 training set (50,000). The x-coordinate in subfigures (a)-(c) is $\mathcal{V}$ in the view of PGD. The x-coordinate in subfigures (d)-(f) is $\mathcal{V}$ in the view of CW. Light yellow and light blue are all instances (50,000) in the view of PGD and CW, while dark yellow and dark blue are top-20% vulnerable instances (10,000) respectively in the view of PGD and CW. Different distributions of dark yellow and dark blue between subfigure (b) and (c) (or between subfigure (e) and (f)) show that, the vulnerable instances in the view of different attacks are not the same.

**Definition 2** (Vulnerability in the view of CW). *In the view of CW, the vulnerability $\mathcal{V}^{\mathrm{CW}}$ regarding $\tilde{\boldsymbol{x}}$ (generated by CW) is defined as*

$$\mathcal{V}^{\mathrm{CW}}_{(\tilde{\boldsymbol{x}}, y)} := \mathrm{z}_y(\tilde{\boldsymbol{x}}) - \max_{i \neq y} \mathrm{z}_i(\tilde{\boldsymbol{x}}), \tag{6}$$

*where $\mathrm{z}_y$ denotes the predicted logits of $\tilde{\boldsymbol{x}}$ (CW) belonging to the true class $y$. The $\max_{i \neq y} \mathrm{z}_i(\tilde{\boldsymbol{x}})$ denotes the maximum predicted logits of $\tilde{\boldsymbol{x}}$ (CW) belonging to the false class $i$ ($i \neq y$).*

As shown in Figure 5, the dark yellow and dark blue are top-20% vulnerable instances in the view of PGD and CW, respectively. The frequency distribution of dark yellow in Figure 5(b) and dark blue in Figure 5(c) is clearly different, and the frequency distribution of dark blue in Figure 5(e) and dark yellow in Figure 5(f) is also different. This phenomenon means that the most vulnerable $10,000$ PGD adversarial variants and the most vulnerable $10,000$ CW adversarial variants are not from the same $10,000$ instances. As a consequence of this inconsistency, if the attack simulated in training is *mis-specified*, the weights of IRAT will be misleading.

## 4   LOCALLY REWEIGHTED ADVERSARIAL TRAINING

To break the limitation of IRAT and train a robust classifier against various attacks, we propose LRAT in this section, for which we perform local reweighting instead of global/no reweighting.

### 4.1   MOTIVATION OF LRAT

**The Reweighting is Beneficial on the Given Attack.** As suggested by GAIRAT (Zhang et al., 2021), the global reweighting indeed improves the robustness when tested on the given attack (PGD) simulated in training. Figures 1(b) and 1(c) also show that, as a global reweighting, GAIRAT improves the robustness against PGD (when PGD is simulated in training). Thus, when training and testing on the same attack, the rationale that we do not need to pay much attention to an already-safe instance under the attack is significant.

**Local Reweighting Can Be Used to Take Care of Various Attacks.** As introduced in Section 3, there is inconsistent vulnerability between instances in different views. Thus, to defend against various attacks, we should perform *local reweighting inside each pair*, while performing *no global reweighting*—the rationale is to fit the instance itself if it is immune to the attack, but not to skip the pair, in order to *passively* defend different attacks in future.

## 4.2 LEARNING OBJECTIVE OF LRAT

Let $\omega(\tilde{\boldsymbol{x}}, y)$ be the weight assignment function on the loss of adversarial variant $\tilde{\boldsymbol{x}}$. The inner optimization for generating $\tilde{\boldsymbol{x}}$ depends on attacks, such as PGD (Eq. (2)). The outer minimization is:

$$\min_{f_\theta \in \mathcal{F}} \frac{1}{n} \sum_{i=1}^{n} \left( \left[ \mathcal{C} - \sum_{j=1}^{m} \omega(\tilde{\boldsymbol{x}}_{ij}, y_i) \right]_+ \ell(f_\theta(\tilde{\boldsymbol{x}}_i^*), y_i) + \sum_{j=1}^{m} \omega(\tilde{\boldsymbol{x}}_{ij}, y_i)\ell(f_\theta(\tilde{\boldsymbol{x}}_{ij}), y_i) \right), \quad (7)$$

where $n$ is the number of instances in one mini-batch; $m$ is the number of used attacks; $\mathcal{C}$ is a constant representing the minimum weight sum of each instance; the notation $[a]_+$ stands for $\max\{a, 0\}$. We impose two constraints on our objective Eq. (7): the first constraint ensures that $\omega(\tilde{x}, y) > 0$ and the second constraint ensures that $\mathcal{C} > 0$. The non-negative coefficient $[\mathcal{C} - \sum_{j=1}^{m} \omega(\tilde{\boldsymbol{x}}_{ij}, y_i)]_+$ assigns some weight to the adversarial data term $\tilde{\boldsymbol{x}}^*$, which serves as a gentle lower bound to avoid discarding instances during training. It can also be seen that different weights are assigned to different adversarial variants, respectively. LRAT *pairs* each instance with its adversarial variants and performs *local reweighting inside each pair*. Figure 4 provides an illustrative schematic of the learning objective of LRAT. We also consider a simple version of LRAT, i.e., *locally simulated adversarial training* (LSAT) that simulating attacks with no reweighting in training that $\omega(\tilde{x}, y) = 1$, $\mathcal{C} < 1$, $m = 1$ and $\tilde{x}$ is generated by any simulated attack, if $\tilde{x}$ is generated by PGD, LRAT recovers the SAT (Madry et al., 2018), which assigns equal weights to the losses of PGD adversarial variant.

## 4.3 REALIZATION OF LRAT

The realization of LRAT consists of the locally simulated and locally reweighted.

**Locally Simulated**. Previous adversarial training methods (Madry et al., 2018; Zhang et al., 2021) only simulated PGD attack in training. As analysed in Section 3, there are inconsistent vulnerablility in different views. Different adversarial attacks simulated in training can also behave differently. For defend against diverse attacks, we should consider simulating diverse attacks in training before reweighting. According to the difference objective function, we simulate two kind of representative attacks that CW attack and *simplified targetted APGD-DLR attack* (SAA).

The loss function in CW is:

$$\mathrm{CW}(f, \boldsymbol{x}, y) := -\mathrm{z}_y(\tilde{\boldsymbol{x}}) + \max_{i \neq y} \mathrm{z}_i(\tilde{\boldsymbol{x}}) - \kappa_c, \quad (8)$$

where $\mathrm{z}_y$ is the predicted logit of $\tilde{\boldsymbol{x}}$ belonging to the true class $y$; $\kappa_c = 50$ is a hyper-parameter to denote the confidence (following (Cai et al., 2018; Zhang et al., 2020a)).

The loss function in targeted APGD-DLR is:

$$\text{Targeted-DLR}(f, \boldsymbol{x}, y, t) = -\frac{\mathrm{z}_y(\boldsymbol{x}') - \mathrm{z}_t(\boldsymbol{x}')}{\mathrm{z}_{\pi_1}(\boldsymbol{x}') - \frac{1}{2} \cdot (\mathrm{z}_{\pi_3}(\boldsymbol{x}') + \mathrm{z}_{\pi_4}(\boldsymbol{x}'))}, \quad (9)$$

where $\pi$ is the ordering of the components of z in decreasing order.

**Locally Reweighted.** The objective in Eq. (7) implies the optimization process of an adversarially robust network, with one step generating adversarial variants from natural counterparts and then reweighting loss on them, and one step minimizing the reweighted loss w.r.t. the model parameters $\theta$. For each variant, the definition of the $\omega$ in Eq. (7) also means a lot. Zhang et al. (2021) heuristically design a non-increasing function $\omega$:

$$\omega(\boldsymbol{x}, y) = \frac{(1 + \tanh(\lambda + 5 \times (1 - 2 \times \kappa(\boldsymbol{x}, y)/\kappa_{\max})))}{2}, \quad (10)$$

where $\kappa/\kappa_{\max} \in [0, 1]$, $\kappa_{\max} \in \mathbb{N}^+$, and $\lambda \in \mathbb{R}$. $\kappa(\boldsymbol{x}, y)$ is the GD defined as the least steps that the PGD method needs to successfully attack the natural instance. $\kappa_{\max}$ is the maximally allowed steps.

---

**Algorithm 1** Locally Reweighted Adversarial Training (LRAT)

**Input:** network architecture parametrized by $\theta$, training dataset $S$, learning rate $\eta$, number of epochs $T$, batch size $n$, number of batches $N$, number of attacks $m$;
**Output:** Adversarial robust network $f_\theta$;
**for** $epoch = 1, 2, \ldots, T$ **do**
  **for** mini-batch = 1,2,…,$N$ **do**
    Sample a mini-batch $\{(\boldsymbol{x}_i, y_i)\}_{i=1}^{n}$ from $S$;
    **for** $i = 1, 2, \ldots, n$ **do**
      **for** $j = 1, 2, \ldots, m$ **do**
        Obtain adversarial data $\tilde{\boldsymbol{x}}_{ij}$ of $\boldsymbol{x}_i$ by the attack $j$;
        Calculate $w_{ij}$ according to $t_{(\tilde{\boldsymbol{x}}_{ij}, y_i)}$ by Eq. (12);
      **end for**
    **end for**
    $\theta \leftarrow \theta - \eta \sum_{i=1}^{n} \nabla_\theta \left[ \left[ \mathcal{C} - \sum_{j=1}^{m} w_{ij} \right]_+ \ell\left(f_\theta\left(\tilde{\boldsymbol{x}}_i^*\right), y_i\right) + \sum_{j=1}^{m} w_{ij} \ell\left(f_\theta\left(\tilde{\boldsymbol{x}}_{ij}\right), y_i\right) \right] / n;$
  **end for**
**end for**

---

However, the efficacy of this heuristic reweighting function is limited. As mentioned in Figure 1 (c), when PGD is simulated in training, the robustness on CW attack decreases. For another, as shown in Figure 6, when CW is simulated in training, this reweighting function shows the same decrease when tested on CW.

Hence, a rational and effective reweighting function is necessary. The generations of adversarial variants follow different rules under different attacks, whereas their common attribute is to mislead the classifier with a low probability of predicting the correct label and a high probability of predicting the false label (Madry et al., 2018; Carlini & Wagner, 2017b). It is reasonable to consider the margin of predicted probability between the correct label and the false label as a variable that affects the reweighting function.

**Definition 3** (The variable factor). *The variable factor $t$ regarding adversarial variant $\tilde{\boldsymbol{x}}$ is defined as*

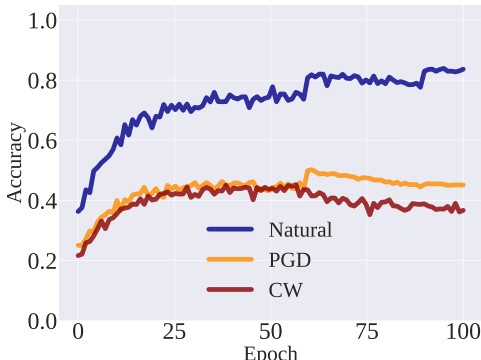

Figure 6: The limitation of Eq. (10). The figure illustrates the performance on the GAIRAT trained model when the CW is simulated in training.

$$t_{(\tilde{\boldsymbol{x}}, y)} := \mathrm{p}_y(\tilde{\boldsymbol{x}}) - \max_{i \neq y} \mathrm{p}_i(\tilde{\boldsymbol{x}}), \tag{11}$$

*where $\mathrm{p}_y$ denotes the softmax on logit of $\tilde{\boldsymbol{x}}$ belonging to the true class $y$.*

Given the variable factor $t$, we also need to consider how to define the reweighting function of $t$. As known, the reason why adversarial training improves adversarial robustness is that adversarial data with a low $t$ are used instead of natural data with a high $t$. It implies the adversarial variant with an high $t$ should be assigned a low weight. Motivated by this thinking, we propose a general reweighting function of $t$ to obtain the weights of the corresponding variants, where the higher variable factor $t$, the lower weights $\omega$.

**Definition 4** (The reweighting function). *The reweighting function $\omega$ regarding the variable factor $t$ is defined as*

$$\omega(t) := \alpha(1 - t)^\beta, \tag{12}$$

*where $\alpha > 0$ and $\beta > 0$ are hyper-parameters of each attack.*

We present our LRAT in Algorithm 1. LRAT uses different given attacks (e.g., PGD, Algorithm 2 in Appendix A) to obtain different adversarial variants, and leverages the *attack-dependent* reweighting strategy for obtaining their corresponding weights. For each mini-batch, LRAT reweights the loss of different adversarial variants according to our reweighting function, and then updates the model parameters by minimizing the sum of the reweighted loss on each instance.

## 5 EXPERIMENTS

In this section, we justify the efficacy of LRAT. We consider $L_\infty$-norm bounded perturbation that $||\tilde{x} - x||_\infty \leq \epsilon$ in both training and evaluations in all experiments. All images of the CIFAR-10 are normalized into [0,1].

### 5.1 EXPERIMENTAL SETUP

We compare our proposed LSAT and LRAT with the no-reweighting strategy (i.e., SAT (Madry et al., 2018)) and previous global-reweighting strategy (i.e., GAIRAT (Zhang et al., 2021)). Rice et al. (2020) show that, unlike standard training, overfitting in robust adversarial training decays test set performance during training. Thus, as suggested by Rice et al. (2020), we compare different methods on the performance of the best checkpoint model (the early stopping model at epoch 60). We employ ResNet (ResNet-18) (He et al., 2016) as the targeted classifier. Our experimental setup follows previous works (Madry et al., 2018; Wang et al., 2020b; Zhang et al., 2021). All networks are trained for 100 epochs using SGD with 0.9 momentum. The initial learning rate is 0.1, divided by 10 at epoch 60 and 90, respectively. The weight decay is 0.0002.

For generating the PGD adversarial data for updating the network, $L_\infty$-norm bounded perturbation $\epsilon_{train} = 8/255$; the maximum PGD step $\kappa_{\max} = 10$; step size $\alpha = \epsilon_{train}/10$. For generating the CW adversarial data (Carlini & Wagner, 2017b), we follow the setup in (Cai et al., 2018; Zhang et al., 2020a), where the confidence $\kappa_c = 50$, and other hyper-parameters are the same as that of PGD above. For generating SAA adversarial data, we follow the adaptive step and DLR loss in (Croce & Hein, 2020), but only choose 3 targeted classes. Those attaining 3 highest scores at the original point $x$ (excluding the correct one), and 20 steps for computational efficiency.

Robustness to adversarial data is the main evaluation indicator in adversarial training (Carlini et al., 2019; Chen et al., 2020; Cohen et al., 2019; Du et al., 2021; Pang et al., 2019; Yang et al., 2021; Zhu et al., 2021). Thus, we evaluate the robust models based on four evaluation metrics, i.e., standard test accuracy on natural data (Natural), robust test accuracy on adversarial data generated by projected gradient descent attack (PGD) (Madry et al., 2018), Carlini and Wagner attack (CW) (Carlini & Wagner, 2017b) and AutoAttack (AA) (Croce & Hein, 2020). In testing, $L_\infty$-norm bounded perturbation $\epsilon_{test} = 8/255$, the maximum PGD step $K = 20$, and step size $\alpha = \epsilon/4$. There is a random start in training and testing, i.e., uniformly random perturbations ($[-\epsilon_{train}, +\epsilon_{train}]$ and $[-\epsilon_{test}, +\epsilon_{test}]$) added to natural instances. Note that, the extended version of GAIRAT that GAIRAT-RST uses additional 500K unlabeled data on CIFAR-10 dataset, and to make a fair comparison, adversarial training methods in our experiments only use original data and do not use additional data.

### 5.2 PERFORMANCE EVALUATION

In our experiments of LRAT, we simulate three representative attacks that PGD, CW and SAA. For adversarial variants CW and SAA, we use our reweighting function (Eq. (12)) to obtain their corresponding weights, where $t$ follows Eq. (11). We choose the three hyper-parameters ($\mathcal{C}$ in Eq. (7), $\alpha, \beta$ in Eq. (12)) that $\alpha = 2$, $\beta = 0.5$, and $\mathcal{C} = 0.1$, and we analyze it in Appendix B.

We evaluate the performance of LSAT first. As shown in Table 1, compared with SAT, by locally simulated on given attack CW and SAA, LSAT performs better when tested on the simulated attack. The results indicate that practitioners should choose to simulate the corresponding attack when there is only one type of attack need to defend against. For adaptive and unseen attacks, the adversarial defense could be described as a barrel effect: if there is a short slab, the barrel would leak. Similarly, if there is a weakness in defending against some attacks, the classifier would fail to predict. In this case, practitioners should choose to simulate the attack with higher intensity (e.g, AA in Table 2). For another, compared to AT, LRAT-SAA performs better on all attacks in testing (PGD, CW and AA), which also indicates that the simulated PGD attack in previous studies can be replaced with SAA to achieve better adversarial robustness.

Then we evaluate the performance of LRAT. As shown in Table 1, compared with LSAT, LRAT can boosts adversarial robustness on the simulated attack even further, which is an evidence that our reweighting function (Eq. (12)) works well. The results show that our locally reweighting strategy is superior to the no-reweighting strategy and can achieve better adversarial robustness on the simulated

Table 1: Test accuracy (%) of LRAT and other methods.

| Methods | Natural | PGD | CW | AA |
|---|---|---|---|---|
| Baselines | | | | |
| SAT | **83.17** | 51.14 | 49.95 | 46.12 |
| GAIRAT | 79.69 | **60.82** | 38.69 | 32.14 |
| LSAT | | | | |
| LSAT-CW | 80.40 | 50.49 | 52.03 | 45.26 |
| LSAT-SAA | 81.58 | 55.04 | 51.84 | 46.95 |
| LRAT | | | | |
| LRAT-CW | 79.65 | 50.01 | **54.27** | 45.09 |
| LRAT-SAA | 81.02 | 54.49 | 51.05 | **47.34** |
| LRAT-SAA+CW | 80.69 | 53.56 | 52.02 | 46.67 |

Table 2: Ablation study.

| | Natural | PGD | CW | AA |
|---|---|---|---|---|
| GD+CW | 79.40 | 58.12 | 42.95 | 36.09 |
| GD+SAA | 80.47 | 58.55 | 41.05 | 40.71 |
| SAA+CW | 80.69 | 53.56 | 52.02 | 46.67 |
| SAA+CW+GD | 80.13 | 57.28 | 45.15 | 38.47 |

attack. It means a lot for the defense on the attack with higher intensity, e,g., LRAT-SAA can perform best on AA. For defending against unseen attacks, we recommend that practitioners simulate multiple attacks and assign some weight to the SAA adversarial data term for each instance during training.

Compared with GAIRAT, LRAT has a great improvement under CW and AA but a decline on PGD. Although GAIRAT improves the performance under PGD and can be a good targeted defense against PGD, its poor performance on CW and AA makes GAIRAT not enough to be an effective adversarial defense in most scenarios . Meanwhile, the ablation study in Table 2 also shows involving geometric distance (GD) as a component leads to the decreased robustness when attacked by CW and AA. In contrast, LRAT reduces the threat of any potential attacks as an effective defense. Thus, the results also show that our local-reweighting strategy is superior to previous global-reweighting strategy. In general, the results affirmatively confirm the efficacy of LRAT.

## 6 CONCLUSION

It has been showing great potential to improve adversarial robustness by reweighting adversarial variants during AT. This paper provides a new perspective to this promising direction and aims to train a robust classifier to defend against various attacks. Our proposal, locally reweighted adversarial training (LRAT), pairs each instance with its adversarial variants and performs local reweighting inside each pair. LRAT will not skip any pairs during adversarial training such that it can passively defend against different attacks in future. Experiments show that LRAT works better than both IRAT (i.e., global reweighting) and the standard AT (i.e., no reweighting) when trained with an attack and tested on different attacks. As a general framework, LRAT provides insights on how to design powerful reweighted adversarial training under any potentially adversarial attacks.

ETHICS STATEMENT

This paper does not raise any ethics concerns. This study does not involve any human subjects, practices to data set releases, potentially harmful insights, methodologies and applications, potential conflicts of interest and sponsorship, discrimination/bias/fairness concerns, privacy and security issues, legal compliance, and research integrity issues.

REPRODUCIBILITY STATEMENT

To ensure the reproducibility of experimental results, we will provide a link for an anonymous repository about the source codes of this paper in the discussion phase.

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

## A  ADVERSARIAL ATTACK

Algorithm 2 and Algorithm 3 are the adversarial data generation of PGD and CW, respectively. The loss function in PGD is:

$$\ell_{CE} := -\log \mathrm{p}_y(\tilde{\boldsymbol{x}}), \tag{13}$$

where $\mathrm{p}_y$ denotes the predicted probability (softmax on logits) of $\tilde{\boldsymbol{x}}$ belonging to the true class $y$.

The loss function in CW is:

$$\ell_{CW} := -\mathrm{z}_y(\tilde{\boldsymbol{x}}) + \max_{i \neq y} \mathrm{z}_i(\tilde{\boldsymbol{x}}) - \kappa_c, \tag{14}$$

where $\mathrm{z}_y$ denotes the logits of $\tilde{\boldsymbol{x}}$ belonging to the true class $y$; $\kappa = 50$ (following the setup in (Cai et al., 2018; Zhang et al., 2020a)) is a hyper-parameter to denote the confidence.

## B  EXPERIMENTAL DETAILS

**Weighting Normalization.** In our experiments, we impose another constraint on our objective Eq. (7):

$$\frac{1}{n}\sum_{i=1}^{n}\left([\mathcal{C} - \sum_{j=1}^{m}\omega(\tilde{\boldsymbol{x}}_{ij}^{*}, y_i)]_{+} + \sum_{j=1}^{m}\omega(\tilde{x}_{ij}, y_i)\right) = 1, \tag{15}$$

to implement a fair comparison with baselines.

**Selection of Hyper-parameters.** It is an open question how to define the decreasing function $\omega$ of $t$ in Eq. (12). (or given the definition of $\omega$ in Eq. (12), how to select the hyper-parameters under different situations.) In our experiments, given an alternative value set $\{0.25, 0.5, 1.5, 2, 4\}$, we choose $\alpha, \beta$ in Eq. (12) that $\alpha = 2, \beta = 0.5$, and given an alternative value set $\{0.1, 0.2, 0.4, 0.6\}$, we choose $\mathcal{C}$ in Eq. (7) that $\mathcal{C} = 0.1$. Our experiments show that the performance has a minor variation between different hyper-parameters, and we choose the optimal hyper-parameters depending on whose robustness attacked by PGD is the best.

**Adversarial Data Term.** To avoid discarding instances during training, we impose the non-negative coefficient $[\mathcal{C} - \sum_{j=1}^{m}\omega(\tilde{\boldsymbol{x}}_{ij}^{*}, y_i)]_{+}$ in Eq. (7) to assign some weight to the adversarial data term. The definition of the adversarial data term can vary as required by different tasks. When practitioners only focus on the robustness under attacks simulated in training, this term can be eliminated. When practitioners focus on the robustness under attacks simulated in training and the accuracy on natural data, this term can be defined as the loss of the natural instance. When practitioners focus on the robustness under attacks both simulated and unseen in training, this term can be defined as the loss of the adversarial variant, and we recommend to use SAA adversarial variant.

## C  EXPERIMENTAL RESOURCES

We implement all methods on Python 3.7 (Pytorch 1.7.1) with an NVIDIA GeForce RTX 3090 GPU with AMD Ryzen Threadripper 3960X 24 Core Processor. The CIFAR-10 dataset can be downloaded via Pytorch. Given the $50,000$ images from the CIFAR-10 training set, we conduct the adversarial training on ResNet-18 for classification. DNNS are trained using SGD with $0.9$ momentum, the initial learning rate of $0.1$ and the batch size of $128$.

## D  DISCUSSIONS ON THE DEFENSE AGAINST UNSEEN ATTACKS

As a general framework, LRAT provides insights on how to design powerful reweighting adversarial training under different adversarial attacks. Due to the inconsistent vulnerability in different views, reweighting adversarial training has the risk of weakening the ability to defend against unseen attacks. Thus, we recommend that practitioners simulate diverse attacks during training. Note that it does not

mean that practitioners should use different attacks indiscriminately—for instance, during standard adversarial training, mixing some weak adversarial data into PGD adversarial data will weaken the robustness on the contrary. The recommended diversity is diverse information focused on during adversarial data generation, such as SAA and CW.

---

**Algorithm 2** Adversarial Data Generation in Projected Gradient Descent Attack (PGD)

---

**Input:** natural data $x \in \mathcal{X}$, label $y \in \mathcal{Y}$, model $f$, loss funciton $\ell_{CE}$, maximum PGD step $K$, perturbation bound $\epsilon$, step size $\alpha$;
**Output:** adversarial data $\tilde{x}$;
$\tilde{x} \leftarrow x$;
**while** $K > 0$ **do**
$\quad \tilde{x} \leftarrow \Pi_{\mathcal{B}_{\epsilon}[x]}(\tilde{x} + \alpha sign(\nabla_{\tilde{x}} \ell_{CE}(f(\tilde{x}), y)))$;
$\quad K \leftarrow K - 1$;
**end while**

---

**Algorithm 3** Adversarial Data Generation in Carlini and Wagner Attack (CW)

---

**Input:** natural data $x \in \mathcal{X}$, label $y \in \mathcal{Y}$, model $f$, loss funciton $\ell_{CW}$, maximum PGD step $K$, perturbation bound $\epsilon$, step size $\alpha$;
**Output:** adversarial data $\tilde{x}$;
$\tilde{x} \leftarrow x$;
**while** $K > 0$ **do**
$\quad \tilde{x} \leftarrow \Pi_{\mathcal{B}_{\epsilon}[x]}(\tilde{x} + \alpha sign(\nabla_{\tilde{x}} \ell_{CW}(f(\tilde{x}), y)))$;
$\quad K \leftarrow K - 1$;
**end while**

---

