# OpenReview forum: "Local Reweighting for Adversarial Training"
_ICLR.cc/2022/Conference — ICLR 2022 Submitted_

### Official Review · Reviewer_vQ9i · 2021-10-31

**Correctness:** 3
**Technical Novelty And Significance:** 2
**Empirical Novelty And Significance:** Not applicable
**Recommendation:** 3
**Confidence:** 4

**Main Review:**

**Strong points**

1. This paper points out the issue of IRAT, which is interesting.

2. This paper is well written and easy to follow.

**Weak points**

The paper claim that IRAT misleads the weights between adversarial variants. However, there is a limited explanation of why IRAT mislead the weights and how local reweighting can prevent those.

There are limited experimental results to verify the efficacy of LRAT.
- [Model capacity] There have not been explored LRAT is effective in large parameter models such as WideResNet34-10.
- [Dataset] There is only a single dataset (CIFAR10) result in the paper. Since the author has shown that there is an inconsistency of strong attacks on different datasets (figure 2), I'm curious how local weights can vary across datasets.
- [Computational efficacy] How long do LRAT-SAA and LRAT-CW take to train compare to SAT?
- [Generality] Does LRAT also can be applied to other adversarial training methods such as TRADES?
- [Limited improvement] I would suggest the authors run the proposed method multiple times and report the variance. The comparison between the gap in average performance and the variance would convince the reader if it is a real improvement.
**Minor points**
- There is a missing explanation of how to rank the attack in Figure 2.

**Summary Of The Paper:**

Authors pointed out that instances-reweighted adversarial training (IRAT) suffer from vulnerability against the unseen types of attacks. Authors claimed that a large number of instances are actually overlooked in IRAT which may cause that phenomenon. Therefore, the authors suggest locally reweighted adversarial training (LRAT) which performs local reweighting between the adversarial variants pair.

**Summary Of The Review:**

Overall, I recommend reject to this paper. I think the contribution is not sufficient for ICLR.

The improvement is limited compared to SAT and LSAT which is train the model with previous attacks such as CW or APGD-DLR. Moreover, there are limited comparisons as I elaborated in the weak points.

However, if all my concerns resolve properly, I will increase my score.

---

### Official Review · Reviewer_AgZp · 2021-10-31

**Correctness:** 2
**Technical Novelty And Significance:** 3
**Empirical Novelty And Significance:** 3
**Recommendation:** 5
**Confidence:** 5

**Main Review:**

Pros
1. The analysis of GAIRAT is thorough and insightful. The paper perform various experiments showing that GAIRAT overfits to PGD and increases the robust accuracy of PGD. However, it is more vulnerable under other attacks like CW and AA.

2. The motivation of using different attacks during training phase is reasonable. As the network overfits the attack in the training phase, diverse attacks make the network robust to wider range of attacks and reduce the overfitting.

Cons.
1. The evaluation of LRAT is incomplete. The paper only evaluates LRAT on CIFAR10 with the ResNet-18. The paper should perform experiments on more datasets such as CIFAR10 or SVHN to demonstrate the effecitveness of LRAT. Besides, WideResnet-34-10 is the most commonly used network architecture for adversarial training, LRAT should also be tested on this network.

2. The motivation of locally reweighting is unclear. It is also shown in the experiment that the performance of LSAT is close to LRAT.

3. The paper is hard to follow as the authors come up with some notation with definition. For example, I cannot understand $\tilde{x}_i^*$ in Eqn 7 and Alg 1.

Minors
1. In Section 4.3, the paper shows that when using GAIRAT with CW attack, the accuracy of CW attack is still lower than PGD. Is it contradictory with analysis in Section 1 and Section 3.


**Summary Of The Paper:**

The paper studies the reweighting of adversarial training. It first points out the problem of GAIRAT: GAIRAT only improves the robust accuracy under PGD and lowers the accuracy of other attacks like AutoAttack. Based on the findings, the authors propose a new method: LRAT, which performs adversarial training using different attacks and gives different weights to the instances generated by different attacks.

**Summary Of The Review:**

The analysis of GAIRAT is thorough and insightful and the motivation of using different attacks is clear. However, the evaluation of LRAT is incomplete and I cannot see the benefit of locally reweighting. Therefore, I tend to reject the paper at current phase.

---

### Official Review · Reviewer_qiUn · 2021-11-02

**Correctness:** 1
**Technical Novelty And Significance:** 1
**Empirical Novelty And Significance:** 2
**Recommendation:** 3
**Confidence:** 4

**Main Review:**

The paper studies the weight strategy of adversarial training. The instance weighting is very old, the readers can not see anything new insight from this paper.  The technique of this paper is very simple without theory analysis. And, the experiments seem that the proposed methods only obtain the marginal gains. The paper is not machine learning paper. I would like to recommend the authors to submit the paper to other application tracks. The details comments are:
1, why instance reweighting is necesary for adversarial training? can you provide some theory insight for this? From the experiments, we can see that the instance reweighting obains some marginal improvements over SAT, but what is the time cost? why not report the time in experiments? If you cost much more time than SAT, but only obtains very little improve on the accuracy, then it is not a smart method.
2, why "the safeness should be attack-dependent"?   can you provide some theory guarantee for this?  what kind of attack make the safeness change? Is there any possibility that different attackes do not change the safeness? how different attack change the safeness? what kind of attack change the safeness towars more safe or unsafe? The main claim of this paper is not rigious at all.
3, Can you prove the proposed algorithm converge? and what is the time cost?
4,Why only consider L_inifity attack? why not try L_1 and L_2 attack? why only ResNet (ResNet-18)? why not try ResNet-34 and ResNet-50 and some other networks? why only use CIFAR-10? why not try imagenet and some other datasets?

**Summary Of The Paper:**

The paper studies the weight strategy of adversarial training. The technical contribution of this paper is too limited, and the experiments show very little improvements.

**Summary Of The Review:**

1, why instance reweighting is necesary for adversarial training? can you provide some theory insight for this? From the experiments, we can see that the instance reweighting obains some marginal improvements over SAT, but what is the time cost? why not report the time in experiments? If you cost much more time than SAT, but only obtains very little improve on the accuracy, then it is not a smart method.
2, why "the safeness should be attack-dependent"?   can you provide some theory guarantee for this?  what kind of attack make the safeness change? Is there any possibility that different attackes do not change the safeness? how different attack change the safeness? what kind of attack change the safeness towars more safe or unsafe? The main claim of this paper is not rigious at all.
3, Can you prove the proposed algorithm converge? and what is the time cost?
4,Why only consider L_inifity attack? why not try L_1 and L_2 attack? why only ResNet (ResNet-18)? why not try ResNet-34 and ResNet-50 and some other networks? why only use CIFAR-10? why not try imagenet and some other datasets?

---

### Official Review · Reviewer_kFgS · 2021-11-03

**Correctness:** 3
**Technical Novelty And Significance:** 3
**Empirical Novelty And Significance:** 3
**Recommendation:** 5
**Confidence:** 4

**Main Review:**

Strengths
1. Weighting different adversarial examples for training is intuitive. And the proposed LRAT addresses some existing problems in IRAT.
2. The proposed method is effective as demonstrated by the experimental results.

Weaknesses
1. The experiments are not convincing enough to demonstrate whether the proposed method is really better than SAT. In table 1, can SAT reach a similar level of robustness at the same (or at least close) performance drop? Namely, what if we slightly increase the PGD step in SAT in tab 1, and the performance on natural example is expected to degrade, while the robustness is expected to be increased. When both SAT and the proposed method degrades into the same (or close) natural example performance, e.g., 83.17 or 81.02, which method provides better robustness?

**Summary Of The Paper:**

This paper presents a new method named LART that assigns weights to adversarial examples during adversarial training for better robustness. Compared to the previous baseline work, i.e., IART, the proposed method has overcame several disadvantages.
Quantitative results demonstrate the effectiveness of the proposed method.

**Summary Of The Review:**

The proposed method is an intuitive improvement on the existing adversarial example weighting method.
However, the experimental results are not convincing enough to reveal the true improvements.

---

### Decision · Program_Chairs · 2022-01-20

**Decision:**

Reject

**Comment:**

Reviewers raised various concerns and authors sent in no rebuttal. In view of the negative consensus, this paper then made a clear rejection case.